

# Numerical Modeling Investigation of Flushing Characteristics and Water Age in a Highly Stratified estuary: Mobile Bay, Alabama, U.S.A.

Gaurav Savant[1], Tate O. McAlpin[1]

[1]U.S. Army Engineer Research and Development Centre, Vicksburg, MS, 39180, U.S.A

*Correspondence to*: Gaurav Savant (Gaurav.Savant@erdc.dren.mil)

**Abstract.** The knowledge of timescales of flushing processes within an estuary is essential for the health, and productivity of the estuary, as well as for optimum estuarine management. The hydrodynamics, flushing times, and freshwater age within Mobile Bay were investigated using a three-dimensional (3D) numerical model, Adaptive Hydraulics. Bay flushing and

freshwater times were analysed for various freshwater flow conditions, wind conditions and the influence of the Coriolis force. The flushing times were directly related to the magnitude of freshwater inputs to the system, with the bay exhibiting an average flushing time of 16.5 days for average river inflows. Freshwater age in the bay was closely associated with the freshwater inflows as well as with location in the Bay and varied from 2 days in the Upper Bay to 21 days in the Lower Bay for average river inflows. Northerly, Easterly, and Westerly winds play an enabling role in the flushing process, with shorter depth-

averaged flushing times compared to those without winds, with Northerly and Easterly winds being the most efficient. Southerly winds increase flushing times by changing the circulation patterns in the Bay.

## 1 Introduction

The time scale of water renewal is an essential indicator of the ecological well-being of a water body such as an estuary, reservoir, or embayment. This water renewal or flushing is of importance to oceanographers because these time scales are an

indicator of the overall robustness of the system, and is indicative of the water quality, pollutant dispersal, and other geochemical processes (Du et al., 2018).

Flushing is a generic term, used to capture the broad characteristics of a system. These characteristics often include water age, exposure time, and flushing time. Deleersnijder et al. (2001) define water age as the time elapsed since a parcel of water entered

the system, exposure time is defined as the total time a tracer or contaminant inhabits a region (Marr 2013), and flushing time is defined as the time a tracer or contaminant stays within the system before being flushed out (Du et al., 2018). Of these three timescales, flushing time is of interest due to its integrative property of representing processes occurring within the system, without explicitly accounting for those processes, and water age is of interest to quantify the spatial variability of local processes (Monsen et al., 2002). Exposure time for a water body includes the re-exposure of the tracer or contaminant to the

water body after leaving the system.

Flushing time, as a water body indicator, came to prominence in the study of reservoirs/lakes and the rate of water exchange. For a well-mixed reservoir, Fischer et al. (1979) represented the flushing time as:

$$\tau = \frac{V_r}{Q} \tag{1}$$


where $\tau$ is the flushing time, $V_r$ is the volume of the reservoir, and $Q$ is the net water input. Because estuaries, bays, etc. have water resources from both freshwater and ocean sources, the simple relationship represented by (1) is modified to account for





freshwater inputs. The flushing time is now related to the time required for the freshwater in the system to be removed from the system. This so-called "*freshwater*" flushing time is represented (Asselin, and Spaulding, 1993; Huang, 2007) as:


$$\tau_f = \frac{V_f}{Q_f} \tag{2}$$

where $\tau_f$ is the freshwater flushing time, $V_f$ is the freshwater volume, and $Q_f$ is the freshwater input to the estuary. In general practice, the freshwater volume and freshwater inflow are averaged over several tidal cycles.

The flushing time, represented by (1) and (2) though useful, is extremely limited and can be of little use in estuarine systems that are impacted not only by tides and freshwater inflows but also by winds and other forcings such as stratification due to baroclinic forces. As a result, several researchers (Du et al., 2018; Huang, 2007; Marr, 2013; Kenov et al., 2012) have attempted to study flushing time using eulerian and lagrangian numerical modeling.

Eulerian models hereafter referred to as numerical models, determine flushing time by simulating a passive tracer and tracking the time for the tracer mass to decrease to a certain threshold. Dyer (1973), Ketchum (1951), and Marr (2013) define this threshold mass as $1/e$ (where $e = 2.71828$) of the initial tracer mass (approximately 37 % of the initial value). The flushing time determined using this method is also known as $e$-folding time. Numerical models vary in complexity from simple batch reactor type models to complex two- and three-dimensional (2D, 3D) models. Batch reactor type numerical models are in essence, numerical representations of Continuously Stirred Tank Reactors (CSTR) and simulate the decay of the tracer by

assuming some decay function *apriori*, these models are of reduced use in estuarine environments because the interplay between the ocean and the river inflows is not a closed system. Estuarine systems are well suited to the application of complex 2D and 3D numerical models, with 2D models being applicable to well-mixed estuaries, and 3D numerical models being applicable to estuaries that experience stratification due to baroclinic, or other forcings. Du et al. (2018), Marr (2013), Choi and Lee (2004), Herman et al. (2007), and Williams (1986) are all examples of numerical modeling of flushing time for various

well mixed as well as stratified estuarine systems.

Water age is a local property of the system, compared to flushing time which is an integrative property of the system. Water age is an indicator of the time elapsed since the water, or a particle, or a tracer entered the system. Numerical models compute the water age of freshwater by analysing two variables, a conservative tracer, $C$, and a water age concentration, $\alpha$. Deleersnijder

et al. (2001) represented the advection-diffusion of these quantities as:

$$\frac{\partial C(t,\vec{x})}{\partial t} + \nabla \cdot \left[ \vec{u} C(t,\vec{x}) - K \nabla C(t,\vec{x}) \right] = 0 \tag{3}$$

$$\frac{\partial \alpha(t,\vec{x})}{\partial t} + \nabla \cdot \left[ \vec{u} \alpha(t,\vec{x}) - K \nabla \alpha(t,\vec{x}) \right] = C(t,\vec{x}) \tag{4}$$

where $K$ is the diffusion tensor, $\vec{x}$ indicates $x, y, z$, and $\vec{u}$ indicates a velocity vector with components in $x, y, z$.


The water age is then computed as:

$$\tau_a = \frac{\alpha(t,\vec{x})}{C(t,\vec{x})} \tag{5}$$

In this paper, we present the validation, and application of the 3D numerical model, Adaptive Hydraulics (AdH), presented in Trahan et al. (2018) and Savant et al. (2018), to the study of flushing times and water age in Mobile Bay, Alabama. The descriptions of techniques and methods presented herein will aid other researchers in the utilization of similar methods to study other strongly stratified microtidal estuarine systems with substantial freshwater river inflows.



## 2 Materials and Methods

### 2.1 Target estuary, Mobile Bay

The Mobile Bay estuary is located in the central portion of the northern coast of the Gulf of Mexico (Figure 1). The Mobile River Basin is the sixth-largest river basin in the United States (Lamb, 1979) encompassing 44,000 square miles in parts of Alabama, Georgia, Mississippi, and Tennessee (Atkins et al. 2004). Mobile Bay is primarily under the influence of a daily astronomical tide with a mean range of 0.4 m, a maximum tropic tide range of 0.8 m and a minimum equatorial tide range of less than 0.1 m (U.S. Department of Commerce, 1990). The primary connections of the bay to the ocean are through Main Pass, and Pass-aux-Herons (Figure 1).

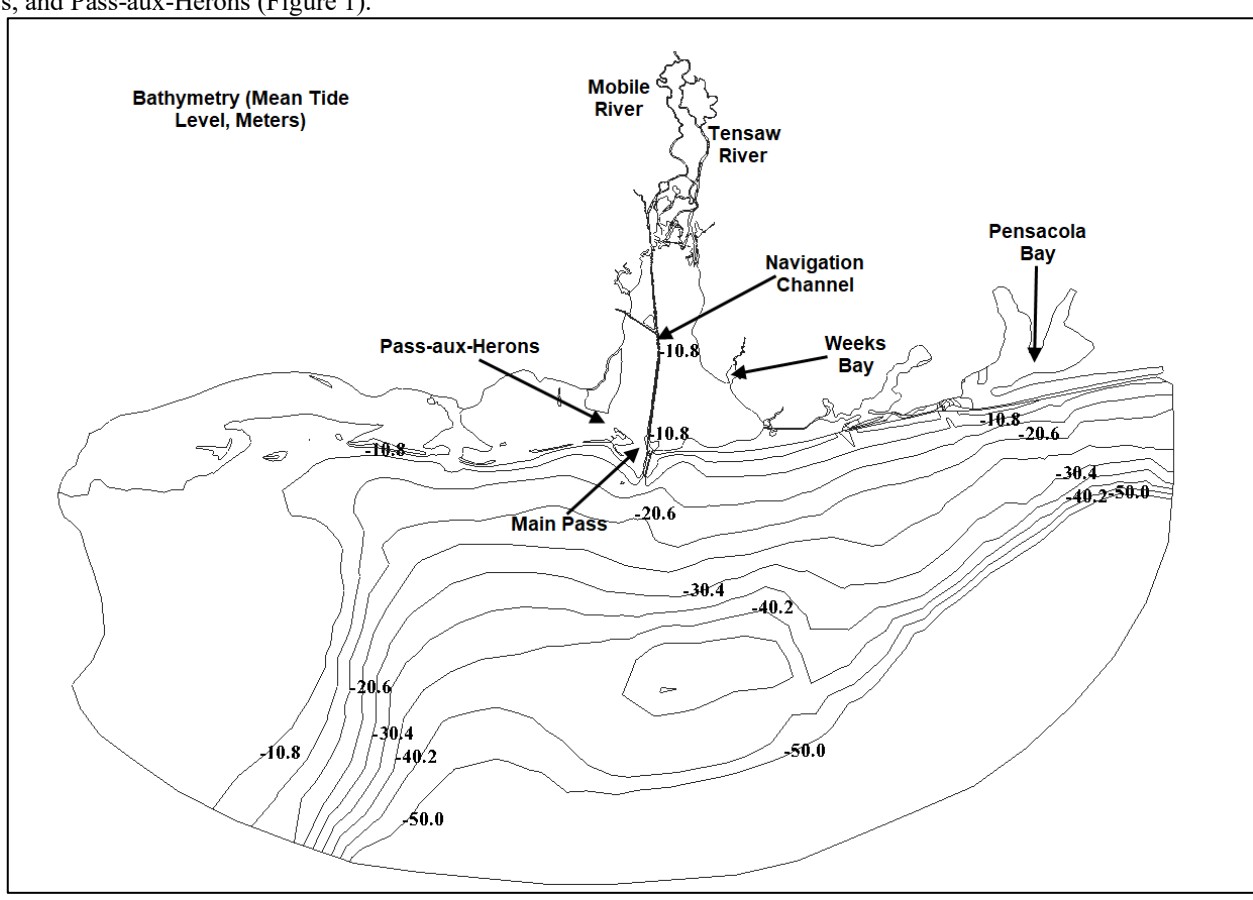

Figure 1: Location Map

The Mobile and Tensaw rivers account for approximately 95% of the freshwater input to the Mobile Bay estuary (Du et al., 2018; Tetra Tech, 2012; Marr, 2013). The average combined discharge of the Mobile and Tensaw River system to the Bay for the period of 1929 to 2010 was 1,848 m³/s but the annual discharge varies considerably from year to year (Schroeder and Wiseman, 1986). The average daily values for the wet and dry seasons are 2,637 m³/sand 802 m³/s, respectively (Marr, 2013; Dinnel et al. 1990, Tetra Tech, 2012). The Mobile Bay system can range from vertically homogenous to highly stratified as





documented by Blumberg et al. (2001), Ryan (1969), Zhao and Chen (2008), and the U.S. Department of Commerce (1990). Salinity stratification predominantly occurs in the 14 m deep navigation channel that cuts through the Main pass with the Gulf of Mexico to the Port of Mobile on the Mobile River (Tetra Tech, 2012). Stratification within the bay is severe during moderate to high river discharge combined with weak winds, and during persistent Northerly winds combined with low freshwater inflows (Schroeder and Wiseman, 1986). Schroeder et al. (1990) reported that during extremely high flows, the entire bay is mainly freshwater and vertically well mixed.

**2.2 Numerical Model**

We use the numerical model, Adaptive Hydraulics (AdH), presented in Savant et al. (2018) and Trahan et al. (2018), to study the flushing times and water age in Mobile Bay.

We, briefly, provide the basic model details in this section, and the interested reader is guided to Trahan et al. (2018) and Savant et al. (2018) for additional details.

AdH is a Finite-Element method (FEM) and implicit time-stepping-based numerical model, that solves the 3D shallow water equations on an adaptive mesh. The basic equations of motion solved by the AdH model are represented as:

$$\frac{\partial u}{\partial x} + \frac{\partial v}{\partial y} + \frac{\partial w}{\partial z} = 0 \tag{6}$$

$$\frac{\partial u}{\partial t} + \frac{\partial uu}{\partial x} + \frac{\partial uv}{\partial y} + \frac{\partial uw}{\partial z} - \frac{1}{\rho_0}\left(\frac{\partial P}{\partial x}\right) - vf - \frac{1}{\rho_0}\left(E_{xx}\frac{\partial^2 u}{\partial x^2} + E_{xy}\frac{\partial^2 u}{\partial y^2} + E_{xz}\frac{\partial^2 u}{\partial z^2}\right) = 0 \tag{7}$$

$$\frac{\partial v}{\partial t} + \frac{\partial uv}{\partial x} + \frac{\partial vv}{\partial y} + \frac{\partial vw}{\partial z} - \frac{1}{\rho_0}\left(\frac{\partial P}{\partial y}\right) + uf - \frac{1}{\rho_0}\left(E_{xy}\frac{\partial^2 v}{\partial x^2} + E_{yy}\frac{\partial^2 v}{\partial y^2} + E_{yz}\frac{\partial^2 v}{\partial z^2}\right) = 0 \tag{8}$$

$$P(z) = P + \int_z^\eta g\rho(z)\,dz \tag{9}$$

where $u$ = $x$-direction velocity; $v$ = $y$-direction velocity; $w$ = $z$-direction velocity; $P$ = pressure; $f$ = Coriolis acceleration; $E_{xx}$ = $x$-direction eddy viscosity; $E_{yy}$ = y-direction eddy viscosity; $E_{xz}$ = *x-direction and z-direction eddy viscosity; $E_{yz}$ = y-direction and z-direction eddy viscosity; $E_{xy}$ = $E_{yx}$* = x-direction and $y$-direction eddy viscosity; g = acceleration due to gravity; $\rho$ = variable density; and $\rho_0$ = reference density.

The transport advection-diffusion is represented by equation (3). The equations of motion and transport are discretized using the FEM in which the variables are represented as linear polynomials on each element. AdH simultaneously solves for the depth using the depth-averaged continuity equation in which two kinematic boundary conditions are used and for the 3D horizontal velocities using the momentum equations. Then a second step computes the 3D vertical velocity using the same discrete continuity equation beginning at either the surface or the bed (the choice is irrelevant as starting from the bottom or the top will arrive at the same results). Temporal discretization in AdH utilizes an implicit backwards Euler scheme represented as:





$$\left(\frac{\partial F}{\partial t}\right)^{t+1} = \beta\left[\frac{\left(\frac{3}{2}F^{t+1} - \frac{1}{2}F^{t}\right) - \left(\frac{3}{2}F^{t} - \frac{1}{2}F^{t-1}\right)}{\Delta t}\right] + \left(1-\beta\right)\left[\frac{\left(F^{t+1} - F^{t}\right)}{\Delta t}\right] \tag{10}$$

where $\beta$ is a factor that varies between 0 (first-order) and 1 (fully second-order), and determines the order of time-stepping, and $F$ is the variable under consideration.

Mass conservation, to machine precision, in AdH was demonstrated in Berger and Howington (2002), and Savant et al. (2018), and the interested user is referred to those publications for details.

## 3 Model Development, and Validation

### 3.1 Meshing, and Bathymetry

The AdH numerical mesh was developed within the Surface Water Modeling System (SMS) software package. The horizontal datum for the AdH mesh was the State Plane Coordinate System (Alabama West, meters) and the vertical datum was mean tide level (MTL, meters).


The model domain extended from Pensacola Bay, FL at the eastern boundary to just west of Gulfport, MS at the western boundary (Figure 1). Bathymetry specified in the Mobile Bay mesh was obtained from three sources. For regions outside of Mobile Bay, depths are based on contour lines and soundings extracted from the United States National Geospatial-Intelligence Agency (NGIA) with vertical datums adjusted to mean-tide-level (MTL, meters). The second source of bathymetric data is a
database generated by the Northern Gulf of Mexico Littoral Initiative (NGLI), a consortium of federal and state agencies, universities, and private contractors. One task performed by the NGLI included bathymetric surveys over portions of the Mississippi Sound. These data were augmented with depths taken from NOS-published charts and then interpolated onto a uniform mesh having 3 arc-second resolutions, in latitude and longitude, to form a bathymetric database. The mesh areas within Mobile Bay were specified with bathymetry data (at a resolution of 0.5m by 0.5m) provided by the USACE Mobile
District (SAM). This consisted of comprehensive surveys of the middle and southern portions of the bay along with surveys of the federally maintained ship channels.

The AdH arbitrary-lagrangian eulerian (ALE) mesh (Trahan et al., 2018; Savant et al., 2018) consisted of layers 1 meter in thickness in the navigation channel, and thinner layers in the shallows surrounding the channel. The mesh generated consisted
of 1,111,383 elements and 231,333 nodes. The lateral element sizes in the base mesh ranged from approximately 60,000 m$^2$ in area (15 km in the longest horizontal length) in the Gulf of Mexico to as small as ~400 m$^2$ in area (20 m in the longest horizontal length) in the navigation channels. The base mesh had maximum of 13 layers in the navigation channel, this resolution reduced to just 1 layer in shallow regions of the Bay that are less that 1m in depth. The open ocean was resolved with 5 vertical layers. AdH is an adaptive model and can refine/unrefine the mesh, horizontally and vertically, as required
based on hydrodynamics and transport (Savant et al., 2018). The model was allowed to refine the horizontal and vertical resolution twice, this can result in a maximum of, if required, $2^n$ layers per vertical layer, where n is the number of refinement times. The horizontal resolution is likewise refined, temporally and spatially, as and when required by the hydrodynamics and transport processes. At this refinement level the model results were converged, i.e., any additional resolution increase did not result in changes to model simulated hydrodynamics and transport. Details of the adaption process are presented in Savant et
al., 2018.

### 3.2 Boundary Conditions, Model Parameters, and Model Spin-Up

The model was validated for 12 months from 1 January 2010 to 31 December 2010. The boundary conditions for this validation exercise included freshwater inflows from the Mobile and Tensaw rivers and tidal influences at the ocean boundary. The model



mesh extended past the Mobile River split into the Mobile and Tensaw rivers (figure 1), and therefore the discharge
observations for the two were combined (figure 2) and applied at the northern extent of the mesh. This was appropriate as
demonstrated by the application of AdH to Mobile Bay by McAlpin (2012). McAlpin (2012) also presented comparisons to
ensure the flow split between the Mobile and Tensaw Rivers was accurately replicated. Since, this flow split is not the primary
focus of this paper and has been published previously, we refer the interested reader to McAlpin (2012) for the flow split
comparisons. Local wind forcings were applied using wind speeds obtained from the United States National Oceanographic
and Atmospheric Administration (NOAA) observations at Ft. Morgan (NOAA station id: 8734673), and Dauphin Island
(NOAA buoy id: DPIA1). The wind speeds were converted to a stress at the water surface using the transformation of Garratt
(1977).

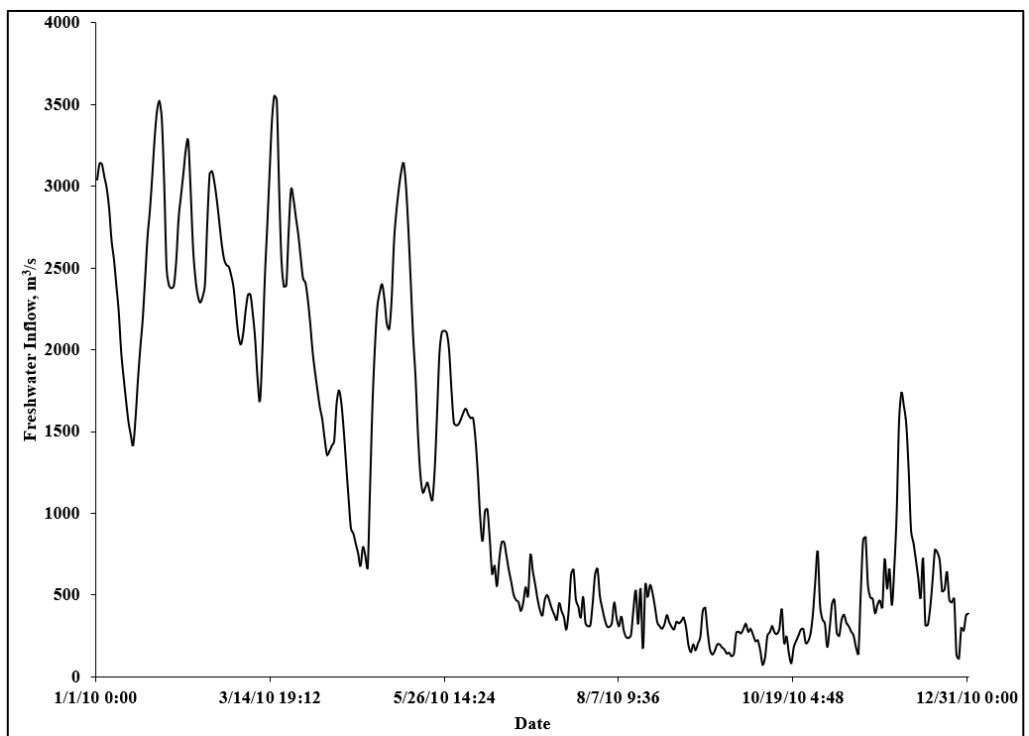

Figure 2: Combined Mobile and Tensaw River Flow in Cubic Meters Per Second. Dates are presented in mm/dd/yy format.

Ocean salinity for the duration of the model run was obtained from the World Ocean Atlas Regional Climatology for the Gulf
of Mexico and was specified as 35.6 ppt. A depth-varying salinity boundary was tested and found to provide no substantial
improvement in the results over the depth-constant salinity.

A spatially and temporally varying tidal elevation ocean boundary was specified using harmonically recreated tides from the
ADCIRC database EC2015 (Szpilka et al., 2016).

The AdH model was executed using a time step of 150 seconds and a second-order backwards Euler time-stepping scheme
(equation 10). The horizontal turbulence was computed using the Smagorinski (1963) formulation, using a Smagorinski
coefficient of 0.2. Following standard practice, horizontal diffusivity was set equal to the horizontal eddy viscosity (Wei et al.,
2016). Umlauf and Burchard (2003) found that the Mellor-Yamada Level 2 , Mellor-Yamada 2.5, k-ε and k-ω produce similar
solutions for the classic entrainment experiment of Kato and Phillips (1969). Though AdH supports several turbulence schemes
such as Mellor-Yamada Level 2 and 2.5, k-ε and k-ω, schemes other than Mellor-Yamada Level-2 require additional transport
equations to simulate the generation and dissipation of turbulence, therefore in the presented numerical model, the vertical





eddy viscosity was specified using Mellor-Yamada level-2, with a Henderson-Sellers (1982) suppression function. Background eddy-diffusivity values are region-specific and must be small enough to represent molecular mixing processes. Etemad-Shahidi and Imberger (2001) observed a mean background vertical eddy diffusivity of $10^{-5} m^2/sec$ in the Swan River estuary, Luketina, and Imberger (1987) observed vertical eddy diffusivity values in the range of $10^{-5} m^2/sec$ to $10^{-9} m^2/sec$ in the Koombana Bay, Li et al. (2005) used values of $10^{-5} m^2/sec$ to $10^{-6} m^2/sec$ to successfully simulate circulation in the Chesapeake Bay. The

background eddy diffusivity for transport was set at $10^{-6} m^2/sec$, this value was intrinsically suppressed, where required, with the Henderson-Sellers formulation as well. Background diffusivity was increased to $10^{-5} m^2/sec$ and decreased $10^{-7} m^2/sec$ to determine the sensitivity of salinity to background diffusivity values. At the increased background diffusivity, the model lost the ability to replicate the vertical distribution of salinity as well as the extents of salinity intrusion into the Bay, and at the reduced background diffusivity values the model was unstable. The Smagorinski coefficient for horizontal turbulence was

perturbed by 10%. In both, increase and decrease, tests the error metrics for water surface elevation comparisons were worse than those obtained using a value of 0.2, except State Docks where there was a marginal improvement with an increased Smagorinski coefficient (RMSE: 0.09, Willmott: 0.95 and correlation: 0.90). Bottom friction was specified using a Manning formulation with a coefficient of 0.017. This value has been used by several researchers (Mcalpin, 2012; Raney and Youngblood, 1982; Webb et al., 2014, Alarcon et al., 2012; Wool et al., 2003) to accurately model hydrodynamics in Mobile

Bay. Therefore, the bottom friction values were not perturbed.

One of the basic tenets, irrespective of the simulation tool utilized, of prognostic numerical simulations of hydrodynamics, is that the initial conditions should not dictate the results of the model simulation; this necessitates the utilization of what is colloquially referred to as the "spin-up" period. Barotropic hydrodynamics, in general, do not require long spin-up periods.

Baroclinic hydrodynamics on the other hand may require several months of spin-up to ensure that the conditions at the start of the period of interest resemble, in general, the field conditions; this is especially true if high fidelity field data are not available to generate initial conditions. The residence or flushing time in Mobile Bay generally varies between 4 and 130 days depending upon freshwater inflows, with some regions (areas on the edges of the Bay and marsh areas) having residence times over 140 days (Marr, 2013). These residence times show that a spin-up period of approximately 4 months will be sufficient to initiate

the production runs. To generate these initial conditions for salinity concentrations and water surface elevations, the AdH model was run for an initial period from 1 September 2009 to 31 December 2009. Similar to the model validation simulations, the spin-up was performed using observed freshwater flows, harmonically recreated tidal elevations, as well as observed meteorological forcings. These hydrodynamic and salinity conditions were used as the initial conditions for all subsequent model simulations.

**3.3 Model Validation**

Mobile Bay is home to the Port of Mobile. The Port of Mobile is ranked 9th in the Unites States of America by tonnage (http://www.asdd.com) resulting in heavy container traffic. Therefore, velocity measurements and persistent salinity observations were not possible. The numerical model was validated through comparisons to water surface elevations at several locations throughout the bay (figure 3-A), and through comparisons to observations of vertical salinity profiles along the

navigation channel (figure 3-B). The model skill in replicating long term salinity behavior was ascertained through comparison to observed salinities over a three-month period between January and the end of May, and one profile in October (figure 6). Model skill in replicating the observations was ascertained through goodness-of-fit analysis using correlation coefficients (Pearson, 1985), Root Mean Squared Errors (Willmott and Matsura, 2006), and Willmott Coefficients (Willmott, 1982).

Figures 4 and 5 graphically illustrate the model skill in capturing the observed water surface elevations and phases, respectively, throughout the bay. Table 1 presents the goodness of fit statistics and indicates that the model is accurately capturing the water surface elevations. The model exhibits accuracy in capturing the water surface elevations as well as the phases of the 6 largest tidal components that constitute the tide in Mobile Bay. The largest errors in the model computed phase were observed in Weeks Bay with a M2 phase error of -6.3%, all other stations had errors smaller than 2% when compared to

those reported by the NOAA.



Discrete vertical profiles of salinity, from January through October 2010 at locations presented in figure 3-B, were available from a field data collection exercise conducted by the University of South Alabama (Dzwonkowski et al., 2011). In most cases model node locations, in horizontal and vertical, did not match the observation location, and therefore the node location closest to the observation location was selected to present these comparisons. For example, if the observation had a z-value of 7 m and the closest model node was at a z-value of 6 m, it was assumed that the z-value of the model is 7 m. This method provided consistency and allowed the avoidance of "tweaking" to use the node with the best agreement to the observations. The AdH computed salinity field was compared to these observations to ascertain model capability in reproducing these vertical salinity profiles. Figure 6 presents this comparison, note that the location 'MB' did not record a complete depth profile of salinity.

Table 2 provides the goodness of fit statistics for the model computed salinity values.

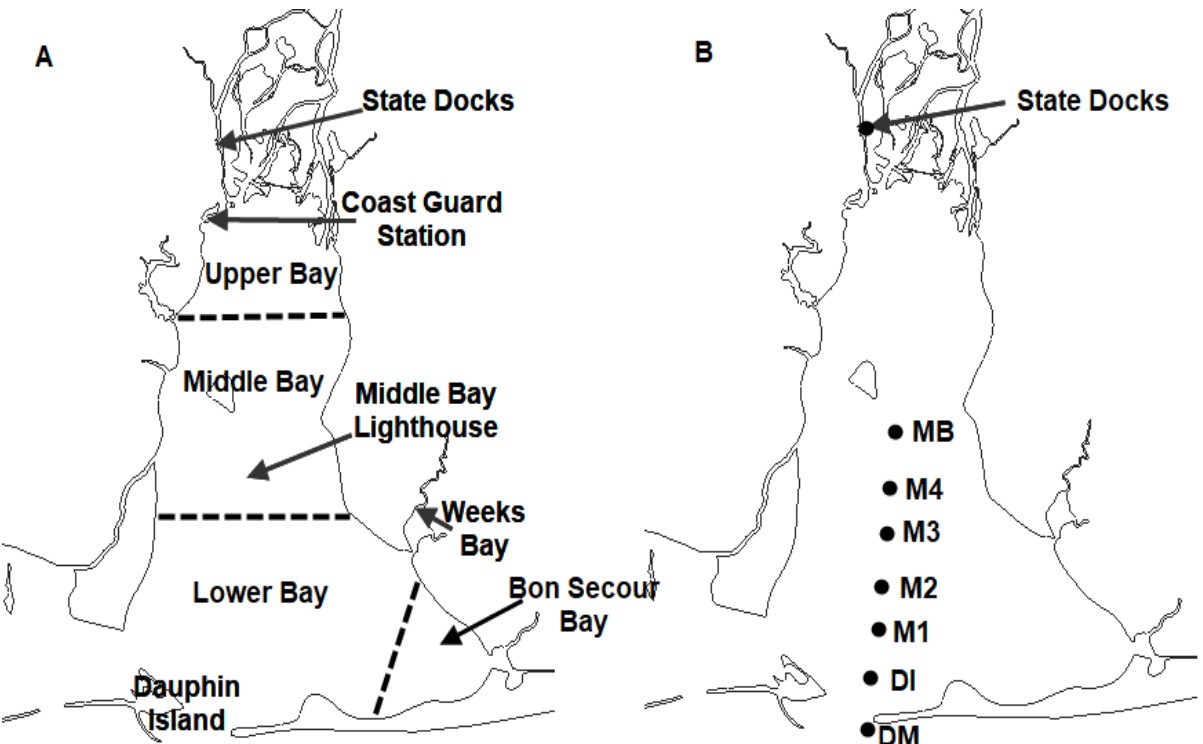

Figure 3: Water Surface and Salinity Observation Locations

To ensure that the model accurately captures salinity behavior over a long period, model results were compared to spot measurements of salinity. For the sake of brevity figure 7 presents the comparison of the model simulated salinity and observed salinity at 4 of the 9 locations (figure 3) between January and June 2010. Table 3 presents the goodness of fit statistics for all stations for the time series salinity comparisons.

Dinnel et al. (1990), Austin (1954), and Marr (2013) have reported that the tidal flow split between the Main Pass and Pass-aux-Herons is 85 and 15% respectively, the presented model computed this split to be approximately 81% and 15%, with the remaining 4% being conveyed through the Gulf Intra-Coastal Waterway (GIWW) (figure 8).

**Table 1. Water Surface Elevation Goodness of Fit Statistics.**



| Location | RMSE, Meters | Willmott Coefficient | Correlation Coefficient |
|---|---|---|---|
| Dauphin Island | 0.09 | 0.92 | 0.86 |
| Weeks Bay | 0.1 | 0.93 | 0.87 |
| Middle Bay | 0.1 | 0.94 | 0.9 |
| Coast Guard | 0.11 | 0.95 | 0.91 |
| State Docks | 0.11 | 0.94 | 0.89 |

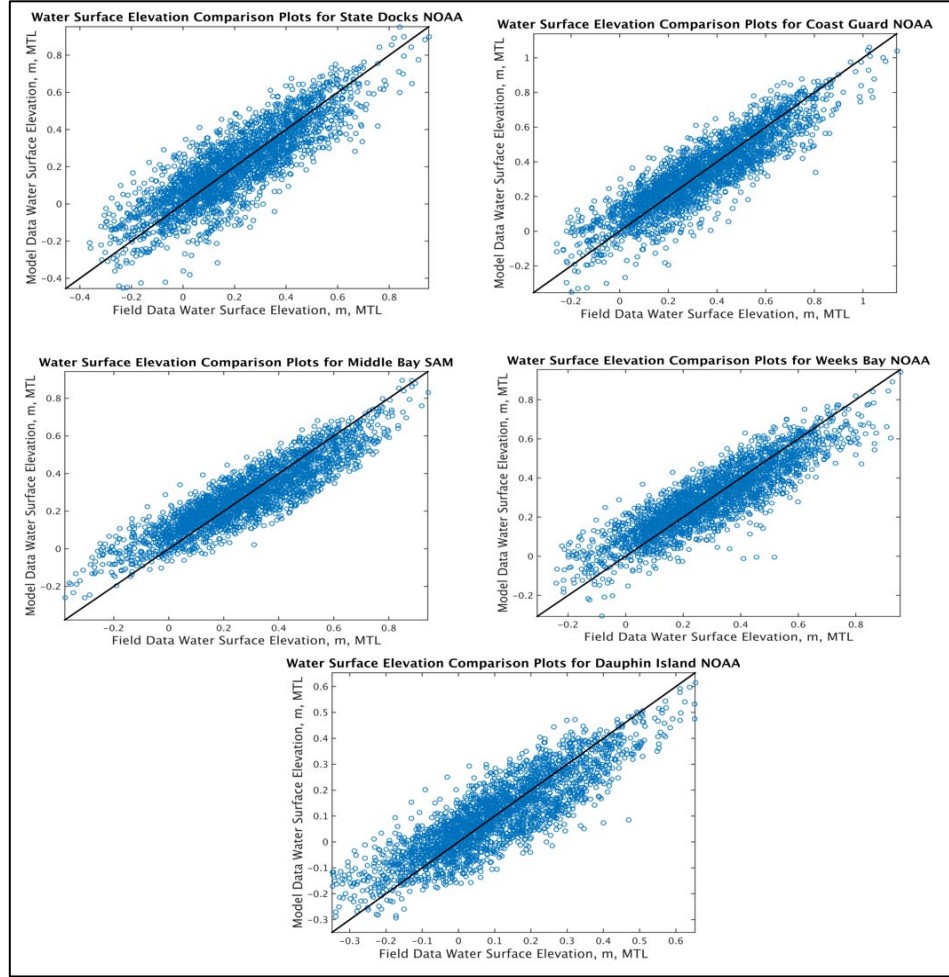

Figure 4: Water Surface Elevation Comparison.

These researchers have also reported a total tidal prism into the bay in the range of 0.45 to 0.41 km³, the presented model computed the average 2010 tidal prism to be 0.42 km³. Figure 9 illustrates the computed residual circulation within the bay. The computed residual currents closely mimic those reported by Byrnes et al. (2013) and provide further confidence that the model is validated to the hydrodynamic conditions within Mobile Bay.



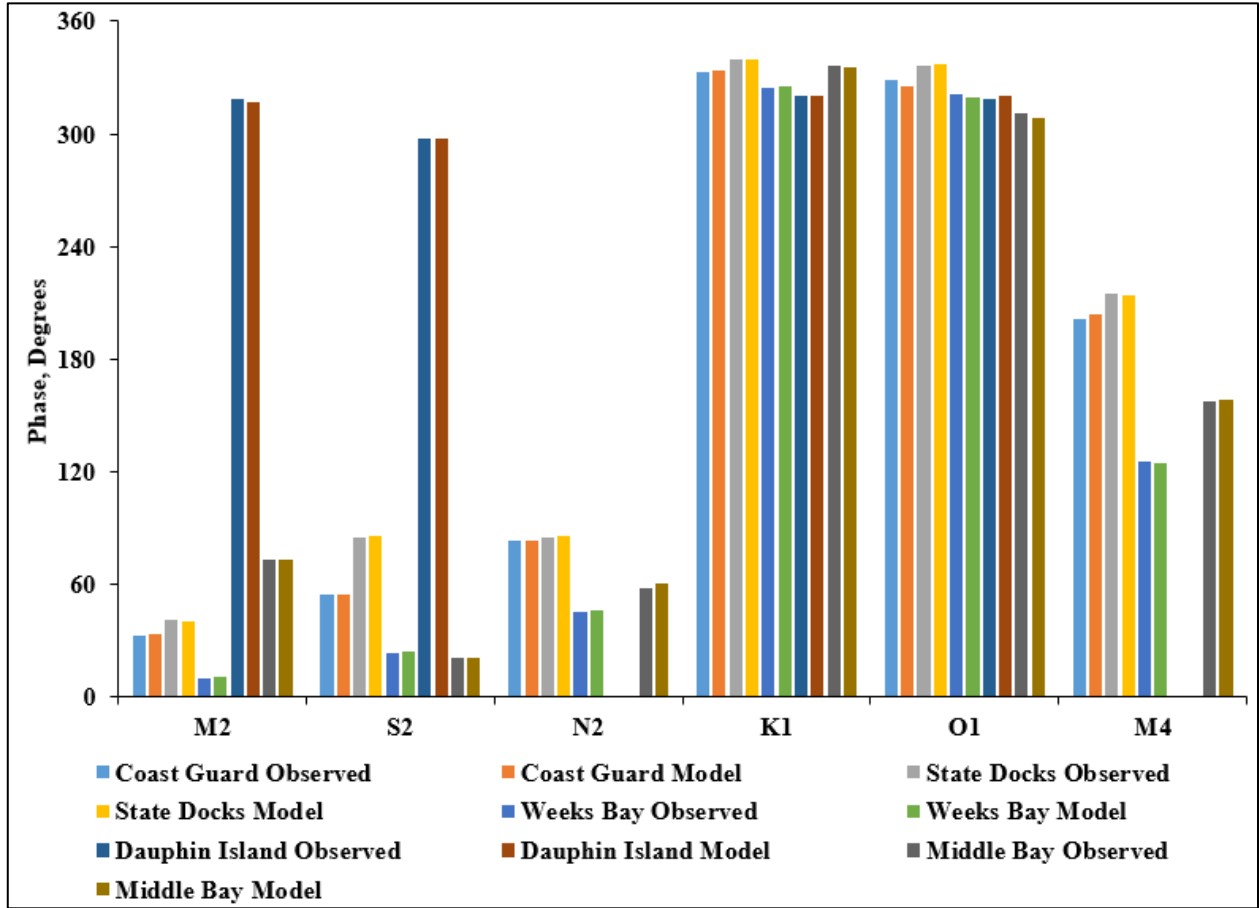

Figure 5: Phase, Degrees. Observed Vs. Model


Velocity observations were not available, therefore a comparison of model velocities to those observed was not possible. However, the skill of the model in replicating water surface elevations, vertical salinity profiles, long term salinity behavior in the bay as well as the accuracy in flow splits provide confidence that the model is validated. State Docks is the upstream limit of salinity intrusion into the Bay, the model accurately replicates the salinity behavior at this location (see figure 7). Salinity
is a function of the velocity behavior, and the accurate representation of salinities further indicates that the hydrodynamics within the bay are being adequately represented.

## 4 Flushing Time, and Water Age

### 4.1 Scenarios Analysed

Flushing time and water age were investigated in a Eulerian manner using the advection-diffusion equation presented
previously in equation (3). A passive dye was released throughout the water column in the bay, with no inflows of the dye from any source into the bay. The simulated scenarios covered the range of freshwater inflows from drought to flood, presented in table 4. The values for the drought, dry, average, wet, and flood flows were obtained from Marr (2013), Tetra Tech (2012), and Schroeder and Wiseman (1986).






Figure 6: Vertical Salinity Profile Comparisons

Mobile Bay is a shallow estuary and wind stresses can play a pivotal role in the flushing characteristics of the system. In the
fashion of Du et al. (2018), winds were applied at a spatially and temporally constant value of 5 m/sec and are tabulated in
table 5.




Similar to the validation simulations, a spatially and temporally varying ocean boundary was specified using harmonically recreated tides from the ADCIRC database EC2015 (Szpilka et al., 2016).

**Table 2. Salinity Goodness of Fit Statistics.**

| Location | Correlation Coefficient | RMSE, ppt | Willmott Coefficient |
|:---:|:---:|:---:|:---:|
| **DM** | 0.99 | 0.72 | 0.98 |
| **DI** | 0.93 | 2.19 | 0.97 |
| **M1** | 0.97 | 2.81 | 0.98 |
| **M2** | 0.94 | 3.02 | 0.95 |
| **M3** | 0.98 | 2.8 | 0.97 |
| **M4** | 0.96 | 2.83 | 0.93 |
| **MB** | 0.95 | 3.25 | 0.91 |


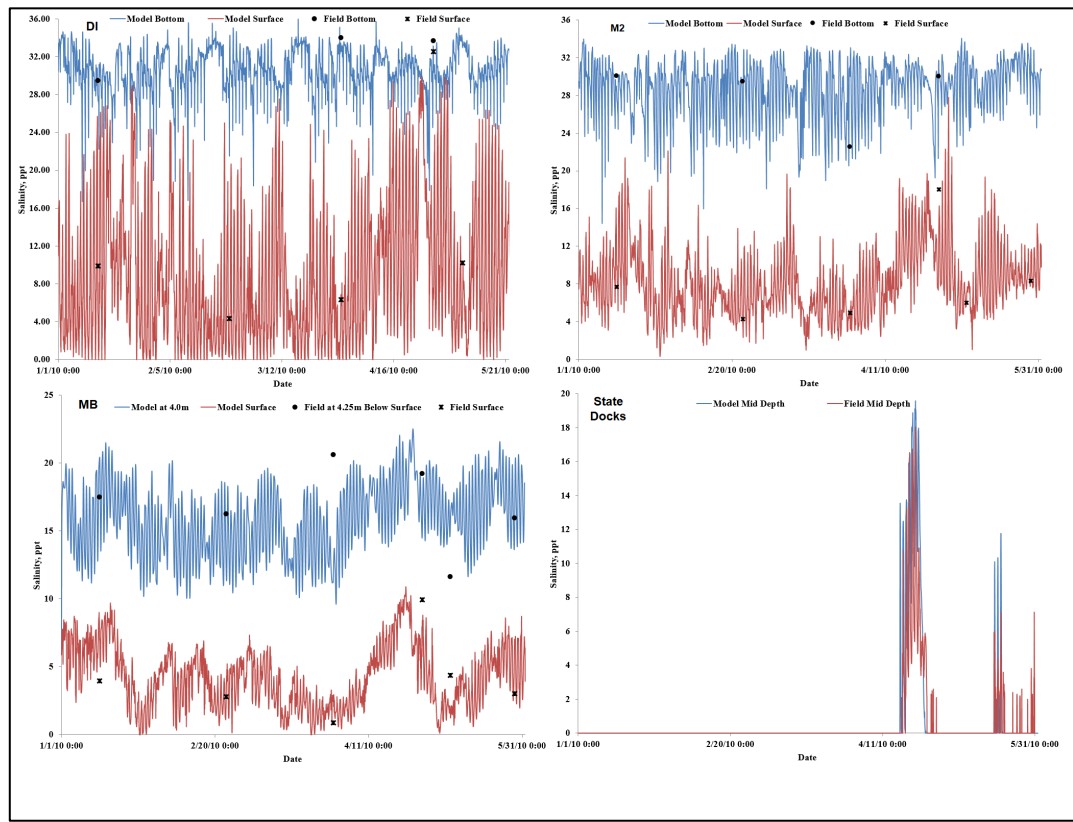

Figure 7: Time Varying Salinity Comparisons




**Table 3. Time Validation For Salinity Goodness of Fit Statistics.**

| Location | Correlation Coefficient | RMSE, ppt | Willmott Coefficient |
|---|---|---|---|
| DM | 0.97 | 1.75 | 0.95 |
| DI | 0.92 | 1.93 | 0.97 |
| M1 | 0.94 | 2.02 | 0.94 |
| M2 | 0.95 | 1.48 | 0.94 |
| M3 | 0.95 | 2.64 | 0.91 |
| M4 | 0.98 | 3.42 | 0.94 |
| MB | 0.94 | 2.78 | 0.89 |
| State Docks | 0.88 | 3.42 | 0.86 |


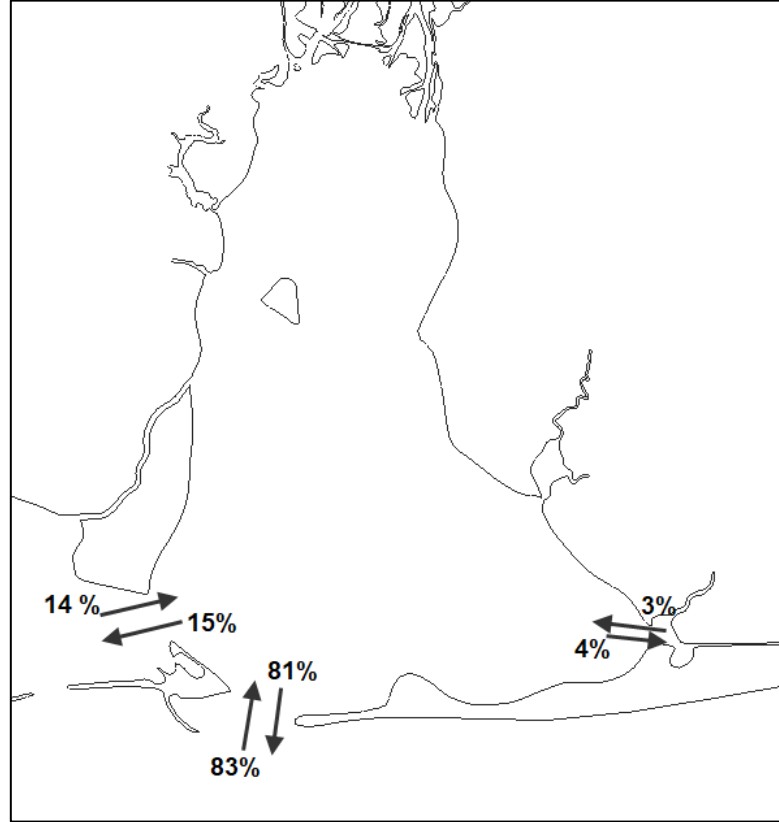

Figure 8: Flow Splits in Mobile Bay.




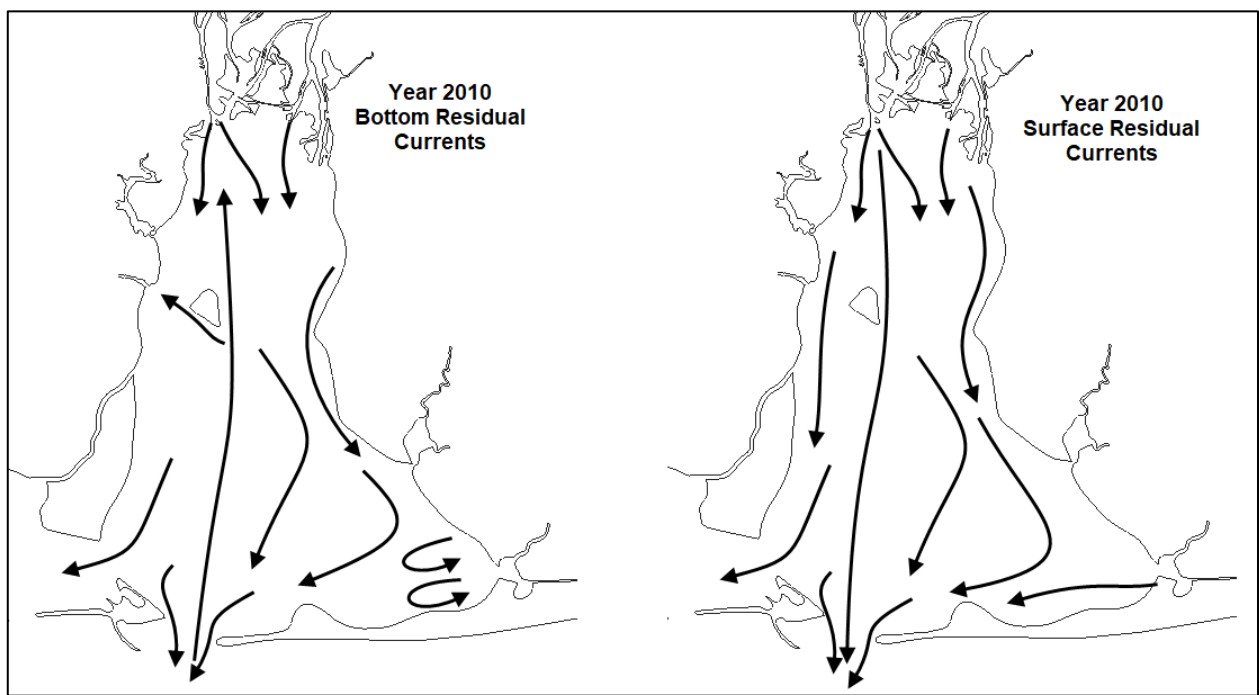

Figure 9: Residual Flow in Mobile Bay.

**Table 4. Numerical Simulation Scenarios.**

| Simulation Number | Flow ($m^3/s$) | Purpose | Tide | Coriolis | Wind |
|---|---|---|---|---|---|
| 1 | Drought (246) | Flushing/Water Age | Astronomical | Yes | No |
| 2 | Dry/Low (802) | Flushing/Water Age | Astronomical | Yes | No |
| 3 | Average (1,848) | Flushing/Water Age | Astronomical | Yes/No | No |
| 4 | Wet/High (2,637) | Flushing/Water Age | Astronomical | Yes | No |
| 5 | Flood(6,747) | Flushing/Water Age | Astronomical | Yes | No |
| 6( a, b, c, d) | Average (1,848) | Flushing/Water Age | Astronomical | No | Yes |


**Table 5. Wind Application.**

| Wind Direction | Speed (m/sec) | Duration (days) | Coriolis |
|---|---|---|---|
| **Easterly** | 5 | 15 | No |
| **Westerly** | 5 | 15 | No |
| **Southerly** | 5 | 15 | No |
| **Northerly** | 5 | 15 | No |





## 4.2 Flushing Time

The numerical model results indicate that the (*e*-folding) flushing times for the system are 105.5, 35, 16.5, 13, and 7.5 days for the drought, dry, average, wet and flood flows, respectively, in the absence of wind stresses and averaged over the water column. The bottom layer (defined as being 2m above the bed in the navigation channel, and 0.5m above the bed elsewhere) flushing times for the same simulations were 111, 36, 17, 14, and 8 days for the drought, dry, average, wet and flood flows, respectively. The surface layer flushing times were 100, 34, 16, 12, and 7 days for the drought, dry, average, wet, and flood seasons, respectively. The flushing times for the bottom layers are consistently longer than that of the surface layers by
approximately 1 to as much as 11 days, indicating an influence of the strong baroclinic stratification, as much as 30 ppt, in the system. The residual circulation in the bottom layers of the navigation channel is upstream and reduces the net outflow of fluid causing an increased flushing time for these bottom layers.

An estimate of the mass fraction of a pollutant remaining in the system is essential for regulatory and ecological management purposes. An analysis of the numerical flushing times was performed to derive empirical relationships between the mass fraction remaining and the flushing time for the pollutant/tracer in the system. For the drought flow (246 m³/s) condition this relationship is:

$$M_f = 1.0e^{-0.009t} \qquad : \text{Bottom}$$
$$M_f = 1.0e^{-0.01t} \qquad : \text{Surface}$$

(11)

Where $M_f$ is the mass fraction remaining after $t$ days.

For a low flow (802 m³/s) this relationship can be expressed as:

$$M_f = 1.0e^{-0.029t} \qquad : \text{Bottom}$$
$$M_f = 1.0e^{-0.03t} \qquad : \text{Surface}$$

(12)

For a wet flow (2,637 m³/s) condition this relationship is:

$$M_f = 1.0e^{-0.076t} \qquad : \text{Bottom}$$
$$M_f = 1.0e^{-0.081t} \qquad : \text{Surface}$$

(13)

For an average flow (1,848 m³/s) condition this relationship is:

$$M_f = 1.0e^{-0.069t} \qquad : \text{Bottom}$$
$$M_f = 1.0e^{-0.067t} \qquad : \text{Surface}$$

(14)

For a flood flow (6,747 m³/s) condition this relationship is:

$$M_f = 1.0e^{-0.12t} \qquad : \text{Bottom}$$
$$M_f = 1.0e^{-0.15t} \qquad : \text{Surface}$$

(15)





Figure 10 illustrates the behaviour of the system in flushing the tracer over time under the flow conditions analysed using the relationships presented. The correlation coefficient, of the presented relations to the model-simulated values, is in the range of 0.92-0.95; and the Root Mean Squared Error (RMSE) is in the range of 0.01 to 0.05.


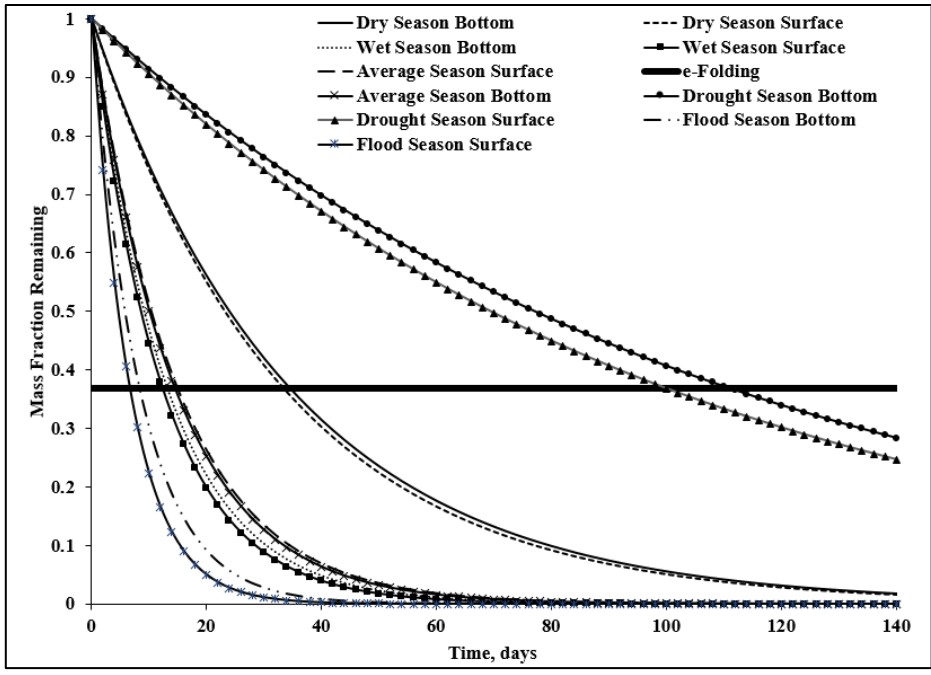

Figure 10: Flushing Time

Mobile Bay is a strongly stratified estuary, and therefore the role played in the flushing of the system by the ocean influx was
investigated. Guo and Lordi (2000), and Du et al. (2018) presented a simple volume conservation equation to determine the volume of ocean flow into a system open to the ocean and receiving freshwater input. This relationship is expressed, in terms of the freshwater inflow into the system, the flushing time, and the volume of the system , as follows:

$$S_o = \left(\frac{V}{T_f}\right) - F_f \tag{16}$$

where $S_o$ is the net influx of water from the ocean, $V$ is the volume of the system, $T_f$ is the flushing time, and $F_f$ is the freshwater inflow. These quantities are illustrated in figure 11.

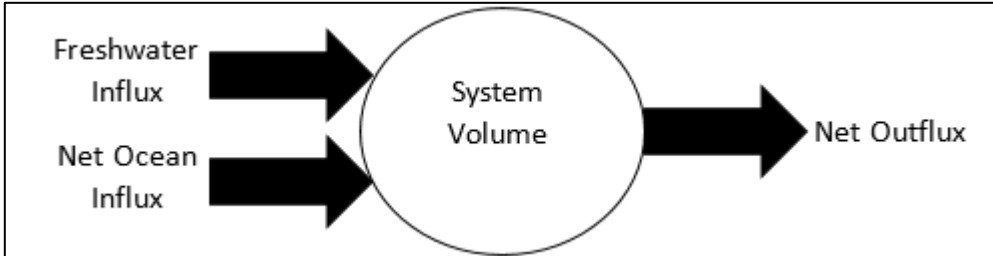

Figure 11: Mass Balance in the System.




Table 6 presents the influence of the ocean influx to the flushing of Mobile Bay.

**Table 6. Ocean Influence on Flushing Time**

| Flow State | System Volume, m³ | $F_f$, m³/s | $T_f$, days | $S_o$, m³/s | Net Outflow, m³/s |
|---|---|---|---|---|---|
| **Drought** | 3.44E+09 | 246 | 105.5 | 1,103.5 | 1,349.5 |
| **Low** | 3.44E+09 | 805 | 35 | 787.6 | 1,592.5 |
| **Average** | 3.44E+09 | 1,848 | 16.5 | 565.0 | 2,413.0 |
| **High** | 3.44E+09 | 2,637 | 13 | 150.98 | 2,787.9 |
| **Flood** | 3.44E+09 | 6,747 | 7.5 | 0 | 6,747 |


The net ocean influx has an inverse correlation to the freshwater inflow into the system, at lower freshwater flows the ocean influx is higher. This behavior is expected, at higher freshwater flows the net force acting against the incoming tide is greater compared to when the freshwater flows are lower. The net ocean influx is primarily generated due to baroclinic flows, and that baroclinic flow due to salinity intrudes into the estuary while acting against the force of the freshwater entering the system as
well as the ebbing of the tide; this again indicates that the greater the freshwater inflow, the lower the net ocean influx. At flood flows, the net influx of water from the ocean is computed to be -1,438 m³/s, this is physically unrealistic, and therefore this value is reported as 0 m³/s in table 6. The authors investigated this relationship, and through a linear regression analysis obtained the following for the range of flows investigated:

$$S_0 = -0.387 F_f + 1184.5 \tag{17}$$


Dyer (1973) presented a simple relation to determine the fraction of an estuary occupied by freshwater. This relationship, based on the ocean salinity (35.6 ppt in the presented paper) and the average salinity in the estuary at flushing time, is presented as:

$$F_{fw} = \left( \frac{\sigma - \overline{S}}{\sigma} \right) \tag{18}$$


Where $F_{fw}$ is the freshwater fraction, $\sigma$ is the salinity of the ocean, and $\overline{S}$ is the salinity in the estuary at flushing time. The $F_{fw}$ computed using equation (16) is 0.2 for drought flow, 0.3 for low flow, 0.53 for average flow, 0.63 for high flows, and 0.9 for flood flows.

River inflows play an important role in the flushing of Mobile Bay (figure 10), therefore an analysis was performed to ascertain any predictive relationship that may exist between the freshwater flows and the flushing times. In addition, Mobile Bay strongly stratifies depending upon river flows causing the relationship between the flow and flushing to be depth dependent. For the surface layers of the bay this relationship, obtained through a power regression analysis, is:

$$T_f = 8376.2 F_f^{-0.819} \tag{19}$$

And, for the bottom layer this relationship is:

$$T_f = 8336.5 F_f^{-0.805} \tag{20}$$

These equations are dissimilar to the ones reported in Marr (2013). Marr (2013) used a depth-averaged barotropic model and reported a flushing time over 120 days at drought flows. This study reports a flushing time of ~105 days for drought flows,
indicating an influence of stratification in the system. At high flows the flushing times reported in this study and Marr (2013) are similar, indicating that at higher flows the estuary is vertically well-mixed.





## 4.3 Water Age

Water age in the system was studied using an Eulerian approach using the water age advection-diffusion in the style of Deleersnijder et al. (2001). A tracer with a boundary concentration of 0.1 kg/m$^3$ was introduced through the freshwater
boundary, this tracer and the consequent computed water age acts as a proxy for the amount of time elapsed since the freshwater entered the system. All water age results are provided 120 days into the simulation.

The freshwater age was analyzed for all scenarios presented in Table 4. The freshwater age exhibited obvious spatial patterns for all flow and wind conditions exhibited. The average upper bay freshwater age was approximately 27, 6, 2, 1, and 0.1 days
for drought, low, average, high, and flood flows, respectively; for the middle bay the ages were 37, 15, 12, 10, and 3 days; and for the lower bay were approximately 58, 34, 21, 17, and 5 days. The deeper navigation channel exhibits greater freshwater ages compared to the shallows surrounding it, this indicates a strong relation of freshwater age to baroclinic influences. The average freshwater age in the navigation channel was 65, 20, 12, 10, and 4 days for drought, low, average, high, and flood flows, respectively. The eastern shore of the bay, consistently, exhibits greater freshwater ages compared to the surrounding
regions. This pattern persists for all flow conditions simulated and indicates a possible area of ecological concern. For drought conditions, the eastern areas of the shallows have a significantly greater freshwater age of > 50 days when compared to the western shallows' freshwater age of approximately 30 days (Figure 12).

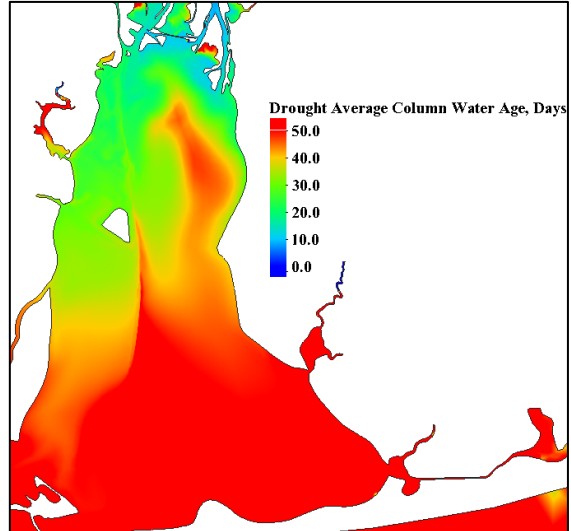

Figure 12: Column average Water Age under Drought Flows

Bon Secour bay, typically, exhibits the highest freshwater age in the range of 50 to 25 days during drought to flood flows, respectively. This is possibly a consequence of the persistent eddy, observed in the residual currents (figure 9), that exists in this area.

The vertical distribution of the freshwater age shows a stratification, in the range of 10 days (figure 13), in the navigation
channel, and this stratification persists for the low as well as the high flow conditions. For the low flow condition, the vertical stratification exists (about < 10 days in the areas surrounding the navigation channel and < 1 day in the shallows) in all regions of the bay. During high flows, the vertical stratification is relegated to the navigation channel, with other areas of the bay being well mixed. High flow simulations show pockets of greater-aged freshwater in Weeks Bay, these are likely a result of several eddies that form in this region because the simulations do not provide any sources of mixing, such as additional freshwater
input. However, this behaviour does provide an indication of Weeks Bay response in case freshwater sources to it are diverted, cut-off, or reduced.



Figure 14 illustrates the representative behaviour, at a transect marking the boundary between the lower and middle bays (figure 3A), of the navigation channel and the surrounding shallows under low and high flows. The navigation channel exhibits
a stratified water age behaviour under both dry and high flows, with the shallows showing stratification for low flow conditions.

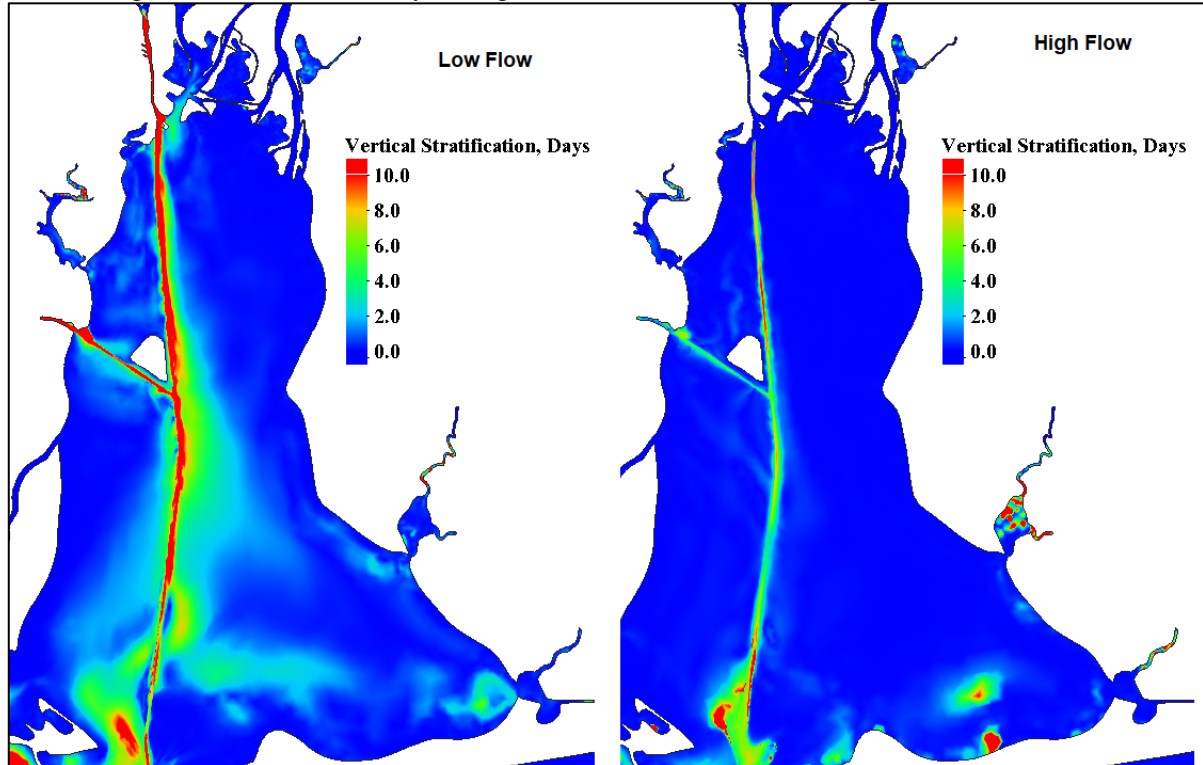

Figure 13: Vertical Stratification of Water Age.

Coriolis forcing appears to play an essential role in the flushing processes in the bay, the column-averaged flushing time of the
bay in the presence of Coriolis force was 16.5 days compared to 14 days in the absence of Coriolis forcings. The primary influence of the Coriolis force is observed in the shallows around the navigation channel (figure 15), with the gradient in water age being sharper near Bon Secour Bay when compared to the behaviour of water age when Coriolis force is on. On the other hand, the western areas of the bay show smearing of the water age when compared to the scenarios when Coriolis was on.



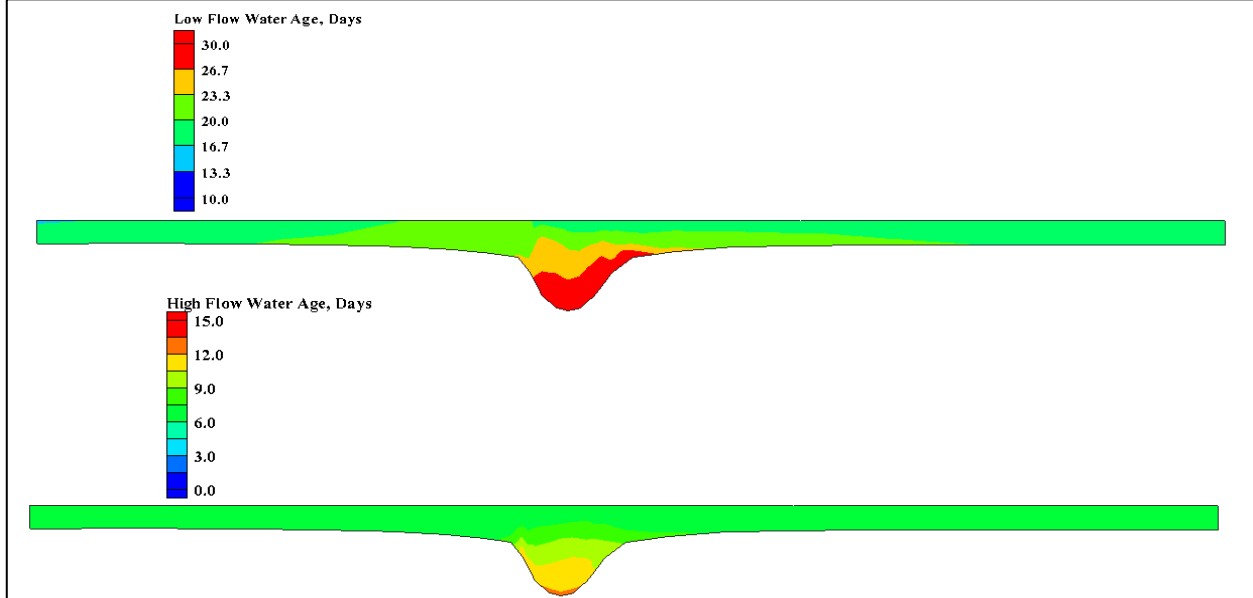

Figure 14: Vertical Stratification of Water Age in the Navigation Channel, and Shallows.

The flushing times in the Bay, for Easterly, Northerly and Westerly winds and an average flow condition, were between 8 to 10.5 days, these indicate a significant reduction in flushing time. Easterly, and Northerly winds experienced the smallest

flushing times of 8 days. Southerly winds had the longest flushing time at 18 days for surface layers and 21 days for bottom layers. Under persistent easterly winds, there is a strong current along the eastern shore directed towards Bon Secour Bay (figure 16) and to the Main Pass, resulting in faster flushing. Under Northerly winds increased flux, through the Main pass (figure 16), enhances flushing.

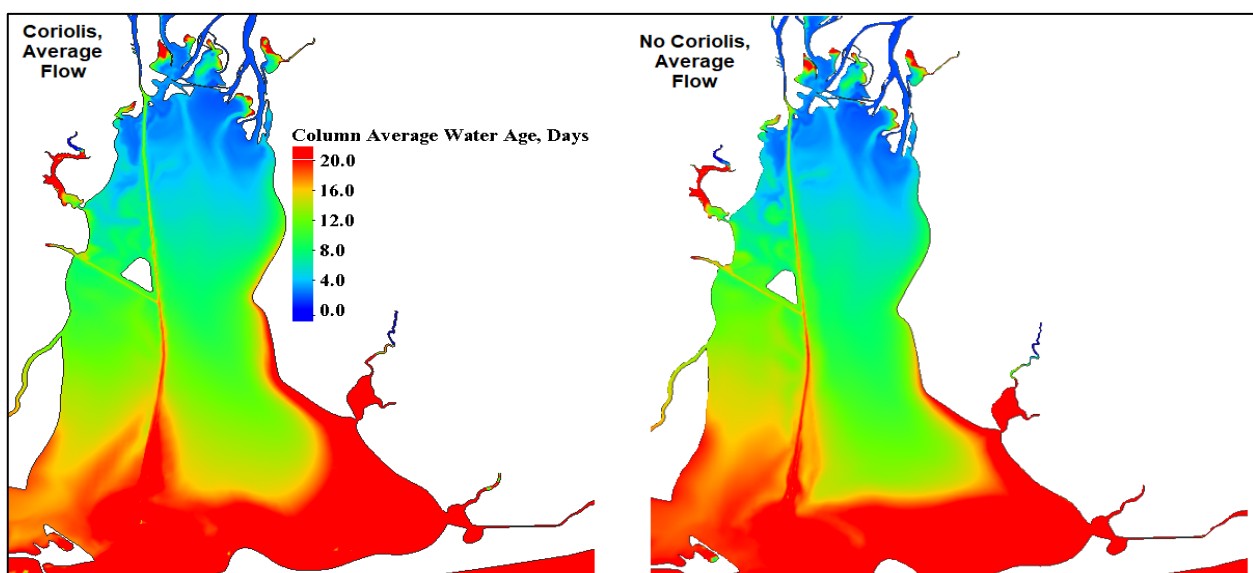

Figure 15: Average Column Water Age With and Without Coriolis.



Under consistent Southerly winds the residual currents in the surface layers of the bay flow northwards. The bottom layers of the bay exhibit a northwards flow trend as well in the lower and middle bay, however the flow in the upper bay exhibits a
southwards residual flow (figure 17).

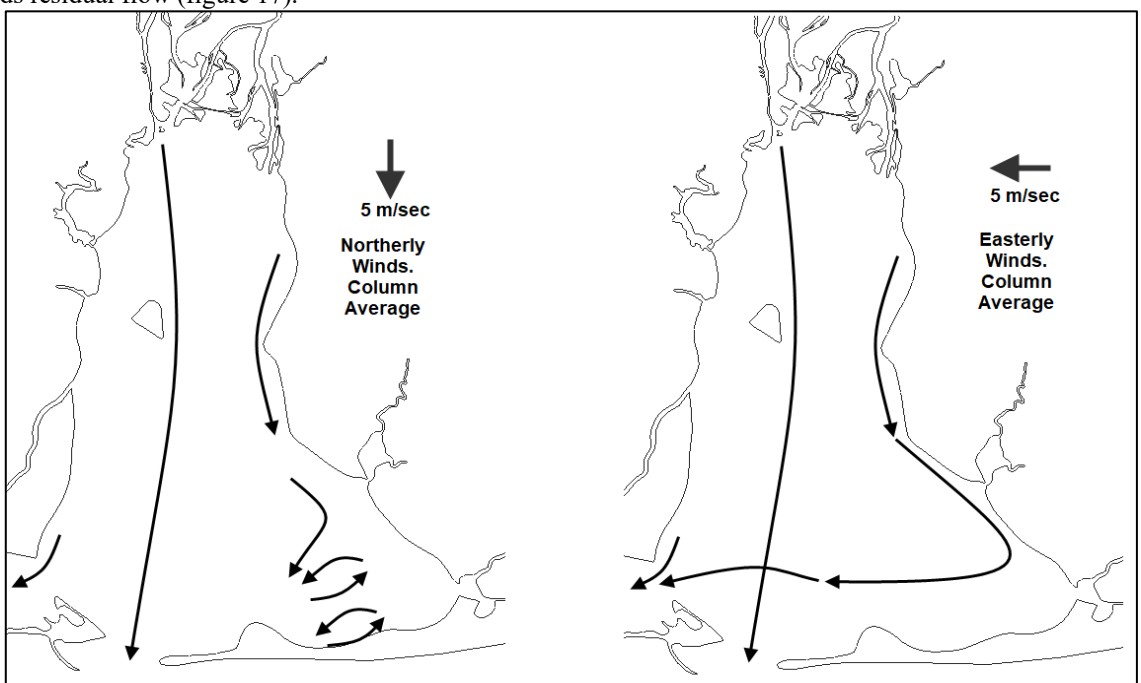

Figure 16: Depth Averaged Residual Circulation with Northerly, and Easterly Winds.



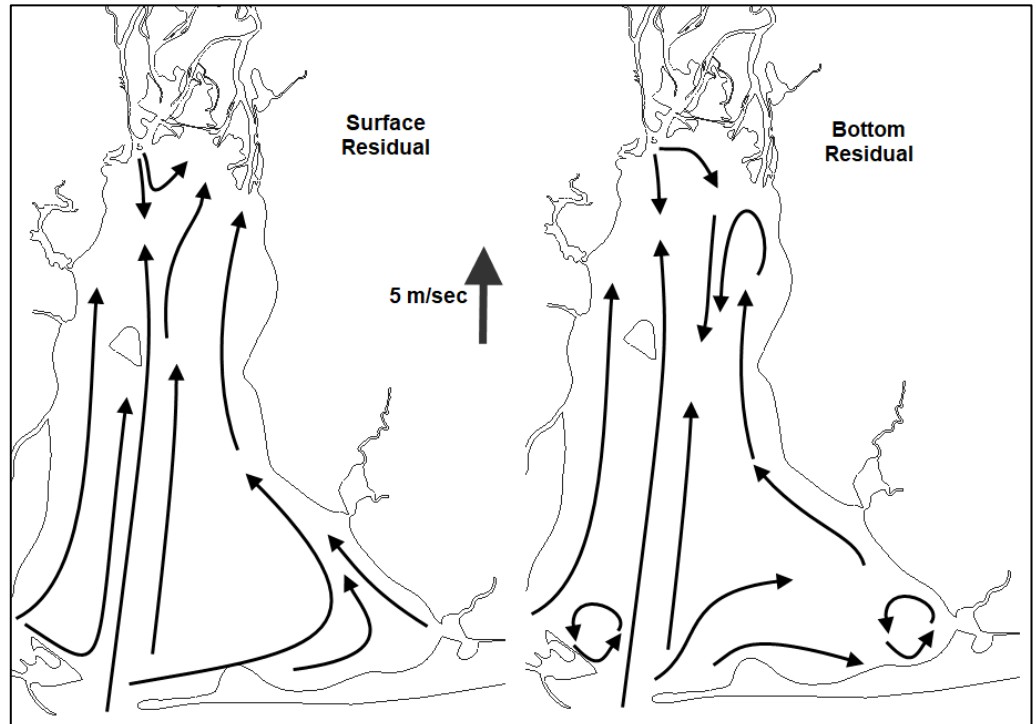

Figure 17: Residual Circulation with Southerly Winds.

## 5 Conclusions

A 3D-AdH numerical model was developed and validated to analyse the flushing times and freshwater age in Mobile Bay. The developed model showed a high degree of skill in replicating observations, as well as the tidal prisms reported in the literature. The model replicated the vertical salinity stratification in the navigation channel, as well as water surface elevations throughout Mobile Bay.

Hydrodynamic modeling in the Mobile Bay estuary indicated freshwater inflows into the Mobile Bay estuary are the predominant factor controlling the flushing and freshwater age in all regions of the estuary. Gravitational circulation due to salinity stratification is an important contributor to the flushing of the estuary and the freshwater age in the navigation channel. An investigation of the flushing response of the bay to varied freshwater inflow revealed a power relationship between the flushing time and the freshwater inflows.

The average flushing time in Mobile Bay spans from 105 days for drought flows to a low of 7.5 days for flood flows. Flushing of the system, for flows less than flood flows, indicates a clear stratified behaviour with the surface layers flushing, on average, 4 days before the bottom layers. This stratification extends to 11 days for drought conditions. For flood flows, the bay is well mixed (except for the navigation channel) and no appreciable influence of stratification, on flushing, is observed. An analysis of the freshwater fraction in the bay for various river inflows showed that the bay ranges from only 20% fresh for drought flows to as much as 90% fresh for flood flows. During flood flows, Bon Secour Bay is the only region of Mobile Bay with significant ocean water. A mass balance analysis indicated that there is an inverse relationship between the freshwater input to the bay and the net ocean influx, and at flood flows there is no net influx from the ocean into the bay.





An analysis of the wind forcing simulation indicated that wind, except Southerly, plays an enabling role in faster flushing of the system. All wind conditions, except Southerly, cause the bay to flush faster than corresponding "no wind" conditions. The average flushing time over Northerly, Easterly, and Westerly wind conditions was approximately 8.75 days, compared to approximately 16 days for 'no wind' conditions. This behaviour is consistent with that reported by Du et al. (2018). Southerly
winds set up a northwards current in the surface and bottom layers and had an average flushing time of approximately 20 days.

The surface layers of the deep navigation channel are the most energetic regions of the bay, whereas Bon Secour Bay is the least energetic. Bon Secour Bay, consistently, exhibits the longest flushing times as well as the oldest freshwater age. This is primarily due to the existence of the persistent eddies in the Bay caused by the geometry. The sideways "*funnel*" shape allows
the formation of two eddies on either side of the GIWW, capturing the simulated tracer, as well as increasing the freshwater age. Bon Secour Bay freshwater age is approximately 50-25 days for drought to flood flows. The freshwater age in Bon Secour Bay is approximately 15 days greater in regions immediately adjacent.

Though the model accurately captures the water surface elevations, vertical salinity profiles, the behaviour of salinities over
time as well as flow splits through the passes, the lack of observed velocity data for model velocity validation is an area that can have an impact on the conclusions presented herein. However, the validation presented throughout the paper provides confidence that the hydrodynamics within the Bay are well captured.

## 6 Code Availability

The AdH code is publicly available through AdH website at https://tinyurl.com/2p9x833x .

## 7 Data Availability

All data used in the presented research is available. The freshwater inflow data is available through https://www.usgs.gov, the ADCIRC database is available at https://adcirc.org. The U.S. Army Engineer Research and Development Centre does not have a data release procedure; therefore, requests are evaluated on an individual basis. Please contact the lead author for access to observed salinity data.
## 8 Author Contributions
The research presented herein has contributions from Dr. Gaurav Savant and Mr. Tate O. McAlpin. Dr. Savant created the mathematical framework for the computations and executed the simulations. Mr. McAlpin created the mesh used in this research.

## 9 Competing Interests

The authors declare that they have no conflict of interest.

## 10 Acknowledgements

The authors acknowledge the Geophysical Computational Modeling project of the U.S. Army Corps of Engineers for funding this research. The authors gratefully acknowledge the constructive suggestions and criticisms from the reviewers.



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
