# Peer review of "Numerical Modeling Investigation of Flushing Characteristics and Water Age in a Highly Stratified estuary: Mobile Bay, Alabama, U.S.A."

_EGUsphere, 2022_

## Author Comment (AC2)

The authors thank the reviewers for their constructive comments on the submitted paper. Through this response we will attempt to answer their comments.

The primary comment made by all reviewers is that the submitted work is similar to that of Du et al., (2018). We present the counterarguments below:

1) Though we use similar methodology presented in Du et al. (2018), the results of the present study are substantially different. Flushing times reported in Du et al. (2018) range between 10 to 30 days for all flows considered, whereas the presented work found that the flushing times range between 7.5 to 105 days for all flows considered. The lowest flows considered in the presented work is 246 m$^3$/s and the highest flow was 6,747 m$^3$/s, and the flushing times were 105.5 days and 7.5 days, respectively. Du et al. considered 280 m$^3$/s as their 10$^{th}$ percentile flow and 5,850 m$^3$/s as their 95$^{th}$ percentile flow, and obtained a flushing time 41.3 and 4.8 days, respectively. At drought and flood flows, our results exhibit greater similarity to the work of Marr (2013). Marr (2013) obtained flushing times using a barotropic model and reported those as being 120 days for drought flows (246 m$^3$/s), and 3.8 days for flood flows (6,747 m$^3$/s). The behavior of the system to persistent forcings such as winds was similar, but again our results in terms of the system response are substantially different.

2) In terms of freshwater age, again the results obtained by the presented study are substantially difference than those presented in Du et al. (2018). An examination of Figure 13 in the present study and Figure 6 in Du et al. (2018) illustrates this dissimilar behavior. The presented results clearly and strongly indicate that the navigation channel governs the behavior of the vertical stratification in freshwater age, whereas the work presented by Du et al. (2018) indicates an equally strong vertical stratification in the shallow as well as the navigation channel with the navigation channel exhibiting little to no stratification in freshwater age at the mouth of the Bay. The presented work indicates a strong stratification behavior throughout the navigation channel.

Even though the general residual behavior of the system is broadly similar, as described above the quantitative behavior between the two studies is substantially dissimilar. The detailed analysis of these differences is beyond the scope of the presented work and these differences can be attributed to several factors including model set-up differences between Du et al. (2018) and the model presented in the work under consideration. The most obvious is the difference in discretization scheme, Finite-difference vs. Finite-Element, respectively. However, this difference should be immaterial or minimal for well-established numerical frameworks. The second difference, and the one more likely to be the cause of the differences is the vertical layering between the two models. Du et al. (2018) represented the domain with a uniform sigma-grid using 5 vertical layers including the deep navigation channel, where as the work under consideration used a variable vertical gridding using an Arbitrary Lagrangian Eulerian (ALE) process. The ALE gridding had 13 layers in the deep navigation channel and the number of layers varied by depth and was increased during run-time dependent upon the stratification being exhibited at a location (refer to section 3.1 for details). This difference in vertical layering can have a substantial impact on the results presented in Du et al. (2018) and the work under consideration here.

3) The work under consideration has also expanded upon the analysis previously presented. We present mathematical descriptions for the various processes including (1) the mass fraction, (2) the freshwater age, and (3) the net ocean influx for the surface as well as the bottom layers of the system. Whereas Du et al. (2018) presented one depth averaged equation for flushing time.

4) The reviewers consider the Coriolis effect minimal; the authors strongly disagree. This study found a difference of ~2.5 days in flushing times for cases when Coriolis was considered versus when Coriolis was not considered. This difference can have dramatic effects on pollutant dispersal as well as pollutant chemical fate.

5) The authors will modify the sentence "Therefore, velocity measurements and persistent salinity observations were not possible." to "Therefore, velocity measurements and persistent salinity observations were not possible within the navigation channel".

The differences we illustrate here provide ample evidence that the science of flushing and other mixing processes in Mobile Bay is unsettled and deserves a robust debate through the publication process.